

# Technical note: An interactive dashboard to facilitate quality control of in-situ atmospheric composition measurements

Yuri Brugnara[1], Martin Steinbacher[1], Simone Baffelli[2], Lukas Emmenegger[1]

[1]Laboratory for Air Pollution / Environmental Technology, Empa, Dübendorf, 8600, Switzerland
[2]Scientific IT, Empa, Dübendorf, 8600, Switzerland

*Correspondence to*: Martin Steinbacher (martin.steinbacher@empa.ch)

**Abstract.** In-situ measurements of trace gases are crucial for monitoring changes in the atmosphere's composition and understanding the underlying processes that drive them. For over three decades, the Global Atmosphere Watch (GAW) programme of the World Meteorological Organization (WMO) has coordinated a network of surface

monitoring stations and facilities with the goal of providing high-quality atmospheric composition measurements worldwide. One of the critical challenges towards this goal is the spatially unbalanced availability of high-quality time series, and the lack of near-realtime quality control (QC) procedures that would allow the prompt detection of unreliable data. Here, we describe an interactive dashboard designed for GAW station operators, but which may be of much wider use, that is able to flag anomalous values in near-realtime or historical data. The dashboard combines

three distinct algorithms that identify anomalous measurements: (i) an outlier detection based on the Subsequence Local Outlier Factor (Sub-LOF) method, (ii) a comparison with numerical forecasts coupled with a machine learning model, and (iii) a Seasonal Autoregressive Integrated Moving Average (SARIMA) regression model. The application, called GAW-QC, can process measurements of methane ($CH_4$), carbon monoxide (CO), carbon dioxide ($CO_2$), and ozone ($O_3$) at hourly resolution, offering multiple statistical and visual aids to help users to identify problematic

data. By enhancing QC capabilities, GAW-QC contributes to the GAW programme's goal of providing reliable atmospheric measurements worldwide.

## 1 Introduction

The Global Atmosphere Watch (GAW) programme, established in 1989 by the World Meteorological Organization (WMO), plays a pivotal role in monitoring and understanding the composition of the Earth's atmosphere (WMO,

2014). This international network of observing stations provides critical data on atmospheric gases, aerosols, and other constituents, which are essential for climate research, environmental policy-making, and public health





assessments. The accuracy and reliability of these measurements are paramount, as they inform scientific models, guide policy decisions, and contribute to our understanding of global environmental changes. GAW data are made publicly available through focus-area specific data centres, where data undergo consistency checks prior to release.

However, data submissions to the data centres are often performed only once a year with delays of several months. Quality control (QC) is a cornerstone of high-quality atmospheric measurements. Ensuring the integrity and precision of data is crucial for maintaining the credibility of scientific findings and the effectiveness of environmental monitoring efforts. QC processes help identify and mitigate errors, biases, and inconsistencies in data collection, thereby enhancing the reliability of atmospheric composition measurements. Effective QC not only improves the

quality of individual measurements, but also strengthens the overall robustness of the GAW programme by ensuring that data from different stations are comparable and consistent. However, achieving comparability and consistency can be challenging and will largely depend on resources, infrastructure, and technical expertise, which vary significantly from country to country. The general lack of near-realtime data sharing often causes instrument malfunctions to go unnoticed for months or even years, resulting in large data losses and a waste of precious

resources.

In recent years, a wide range of powerful tools for enhancing QC processes has emerged in the field of data science. The rise of big data and advanced analytical techniques has revolutionized our way to handle and interpret large amounts of atmospheric data. One particularly relevant area within data science is anomaly detection in time series, which focuses on identifying unusual patterns or outliers within a sequence of data points collected over time (see,

e.g., Schmidl et al., 2022). In the context of atmospheric sciences, anomaly detection can help pinpoint measurement errors, instrument malfunctions, or unexpected atmospheric events that require further investigation (e.g., El Yazidi et al., 2018; Barré et al., 2021; Resovsky et al., 2021; Cristofanelli et al., 2023).

This paper introduces an interactive dashboard designed to facilitate quality control of in-situ atmospheric composition measurements, with a particular focus on recent measurements (i.e., last few months) but applicable

to any period since 2015. The dashboard integrates anomaly detection algorithms and makes use of near-realtime numerical forecasts by the Copernicus Atmosphere Monitoring Service (CAMS). By providing a user-friendly interface that visualizes data and highlights potential anomalies, the dashboard enables researchers and technicians to rapidly identify and address irregularities. However, GAW-QC is intended as a guidance to expert decision, and the ultimate task of flagging the data remains with the station operator, who has the best local knowledge. This



innovative tool improves the accuracy of atmospheric measurements and supports the GAW programme by ensuring that the collected data are of the highest possible quality.

## 2 Data sources

We use historical data of $CH_4$, CO, $CO_2$ and $O_3$ at hourly resolution measured at GAW stations, which are available at the World Data Centre for Greenhouse Gases (WDCGG) and the World Data Centre for Reactive Gases (WDCRG).

We chose these four gas species because they are the most commonly measured at GAW stations. At the time of writing (October 2024), 94 stations were supported by the GAW-QC application (Fig. 1; Table S1). These are the stations that submitted data to the world data centres at least once since 2021.

The archive of numerical forecasts is obtained from CAMS (Peuch et al., 2022). The CAMS Global Atmospheric Composition Forecasts (hereafter just CAMS forecasts) have a horizontal grid resolution of ca. 40 km, a time

resolution between 1-3 hours (depending on the variable) with two analyses per day (at 00 and 12 UTC; hence, we use forecasts with up to 11 hours lead time). They are produced using ECMWF's Integrated Forecasting System (IFS) model with additional modules enabled for aerosols, reactive gases and greenhouse gases, and with the assimilation of additional satellite measurements (Eskes et al., 2024; CAMS, 2024). These forecasts are independent from in-situ measurements.


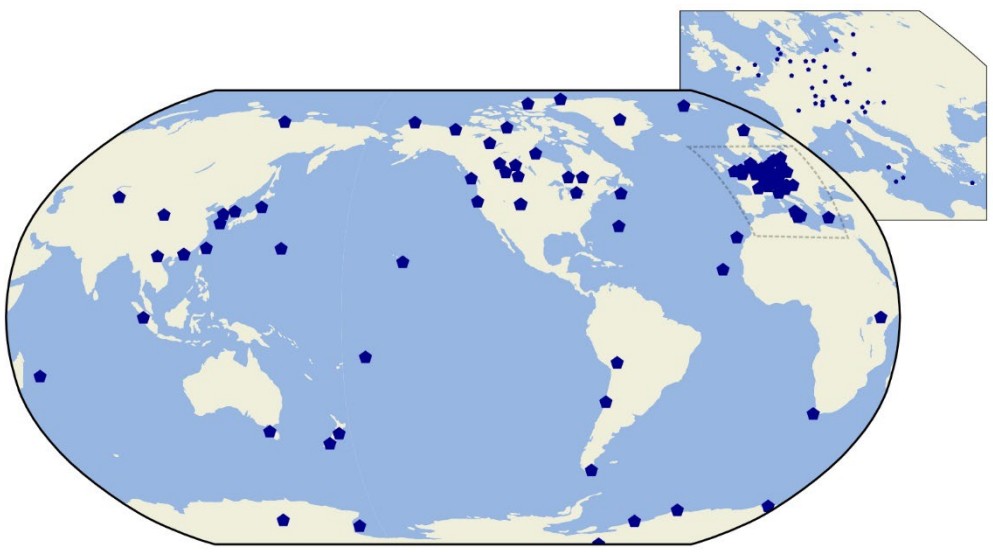

**Figure 1: Map of the 94 GAW stations that were supported by GAW-QC at the time of writing.**



**Table 1: Variables extracted from the CAMS forecasts**

| Variable | Abbreviation | Level | Resolution |
|---|---|---|---|
| $CH_4$ / CO / $O_3$ | ch4 / co / o3 | 3D | 3-hourly |
| $CH_4$ / CO / $O_3$ | tc_ch4 / tcco / gtco3 | total column | hourly |
| Black carbon (550nm o.d.) | bcod | total column | hourly |
| PM10 | pm10 | surface | hourly |
| Water vapour | tcwv | total column | hourly |
| Wind ($u$ component) | u10 | 10 m | hourly |
| Wind ($v$ component) | v10 | 10 m | hourly |
| Temperature | t2m | 2 m | hourly |
| Pressure | msl | mean sea level | hourly |


The IFS model is upgraded regularly, which poses a challenge when using the forecasts to train a machine learning model. In fact, we did not consider data before 2020 because of a change in the vertical resolution that took place in July 2019. A better alternative to an archive of forecasts would be a reanalysis (e.g., Inness et al., 2019); however, no such product is currently available in near-realtime for atmospheric composition variables.

The CAMS forecasts provide data on over 50 chemical species and seven different types of aerosol, in addition to several meteorological variables. We use only a small part of these data: Table 1 lists the variables that we extract from the CAMS forecasts and feed to the machine learning model (note that $CO_2$ is not available). For 3D variables, we use the following pressure levels: 1000 hPa, 950 hPa, 925 hPa, 900 hPa, 850 hPa, 700 hPa and 600 hPa. The selection criteria for these variables are described in Sect. 3.2.

In addition, we use data from atmospheric stations of the Integrated Carbon Observation System (ICOS) (Heiskanen et al., 2022), for validation purposes. By providing both near-realtime (L1, i.e. automatically processed data without any further screening and flagging by the operators, see Hazan et al., 2016) and quality controlled (L2) data, ICOS constitutes an almost ideal validation data set. The quality control in ICOS follows various manual and semi-automatic procedures performed by trained scientists, usually on the sub-hourly scale (see, e.g., Cristofanelli et al.,

2023). We consider a data point (hourly mean) "flagged" by the CAMS operators if one of the following conditions is met: (i) more than 50% of measurements contributing to the hourly mean were removed in the L2 data; or (ii) the absolute difference between L1 and L2 exceeds 5% of the L1 value. For stations with towers, we only use the data from the highest sampling height. Note that ICOS data do not include $O_3$.

The main caveat of our validation strategy is that several ICOS station operators have a tendency to overflag, meaning that for practical reasons entire days are removed from L2 data, even though only few hours may have been of low quality. As a mitigation measure, we exclude from the data set periods longer than 7 days in which all data points had been flagged. Another limitation of the ICOS data set is its geographical representativeness, since nearly all ICOS stations are located in Europe.

## 3 Methods

GAW-QC is implemented in Python language using the Dash framework (https://dash.plotly.com). Anomaly detection in GAW-QC is based on three distinct methods, each focusing on a specific time scale and type of anomaly. (i) A Subsequence Local Outlier Factor (Sub-LOF) algorithm detects anomalous sequences of measurements on the scale of a few hours; it is particularly suited for isolated outliers and changes in variability. (ii) CAMS forecasts combined with machine learning provide predictions at hourly resolution, allowing the detection of systematic biases on most time scales (down to a few days). (iii) a Seasonal Autoregressive Integrated Moving Average (SARIMA) regression model predicts monthly mean values based on previous data and can highlight outliers at the monthly scale.

These methods were chosen based on performance, efficiency and ability to work well with little training data. The need for near-realtime QC implies that we could not use data from nearby stations (which are generally not available in near-realtime), nor other variables measured at the target station (to avoid a too large burden on the user and too complex data format requirements). In the following sections, we describe each of the three methods in details.

### 3.1 Outlier detection with Sub-LOF

LOF (Breunig et al., 2000) is an unsupervised, distance-based algorithm that produces an anomaly score for each data point (the higher the score, the more "anomalous" the point). Sub-LOF is its extension to time series and produces an anomaly score for each sub-sequence of length $n_s$ within the series. We use the Sub-LOF implementation provided by Schmidl et al. (2022), in which an additional step assigns an anomaly score to each data point as the average of the anomaly scores of the sub-sequences that include the target data point. Despite its relative simplicity, Schmidl et al. (2022) have shown that Sub-LOF is among the best performing anomaly detection algorithms for a large collection of data sets.



LOF anomaly scores are based on a subsample of $k$ nearest neighbours (hence the "local" in the name). For a time series, this means that, e.g., seasonal differences can be taken into account without any pre-processing of the data. To find suitable hyper-parameters $n_s$ and $k$, we run a grid search (that is, we tested the algorithm with many different combinations of hyper-parameters) using data from ICOS stations that have at least 30 flagged measurements (i.e., the anomalies that we want to detect) between April 2021 and March 2024. This led to $n_s$=3 and $k$=100, a choice

that we further discuss in Sect. 4.

Being an unsupervised method, Sub-LOF does not need a training data set and, therefore, does not need historical data to work. However, we take advantage of historical data to increase the sample size (which, ideally, should be much larger than $k$) and to define station-tailored thresholds for the anomaly score that GAW-QC uses to flag anomalous data points.

The anomaly score of a point can be calculated only if there are at least two ($n_s$ − 1) valid measurements next to it (i.e., it must be part of at least one valid 3-hour sequence). Therefore, in case of frequent missing values, it might not be possible to determine if certain measurements are anomalous. To partially compensate for this limitation, we implemented a simple additional test that flags all values exceeding the historical highest or lowest value by half the historical range.

**3.2 Downscaling of CAMS forecasts**

The CAMS forecasts have a too coarse resolution to well reproduce the atmospheric composition at most stations. Moreover, they are often affected by systematic biases. To deal with these limitations, we train a machine learning model for each station using the archive of forecasts for the most representative grid point and the historical measurements at the station, in order to obtain a more reliable "downscaled" point forecast (hereafter CAMS+). A

similar approach was followed by Bertrand et al. (2023) to improve CAMS forecasts of aerosols.

We use the Extra-Trees (ET) model (Geurts et al., 2006), a variation of the random forest (RF) model with randomized tree nodes splits, as implemented in the Python library scikit-learn. This model has two important advantages with respect to a standard RF: (i) it strongly reduces overfitting without the need of complex hyper-parametric choices, hence it has better chances to work well for different stations and gas species using the same hyper-parameters;

and (ii) it is much cheaper to train, because it does not need to find the best splits.

By training both ET and RF models using GAW stations that had at least 50% data availability during 2020-2022 (the training period) we found that the ET model outperformed RF and other classic machine learning models on



average for all tested gas species and for different validation scores for the test period 2023 (Figure S1). Moreover, by reducing the size of the training period, we saw an increase of the performance difference, implying that ET is
less sensitive to the size of the training data set (Figure S2). The (station- and gas-independent) hyper-parameters of both models were optimized through the same grid search procedure, which used a random train/test split of 75%/25% over the 2020-2022 period, different for each station and gas species.

The ET model is based on 200 trees with maximum tree depth of 15 layers and minimum sample size for splitting a node ($n_{min}$) of 20 elements. $n_{min}$ is particularly important as it has the effect of smoothing the results. Our choice
of a relatively high $n_{min}$ is justified by the noisiness of atmospheric measurements and helps to reduce overfitting. On the other hand, we anticipate the model to underestimate extreme events because of the large smoothing.

The most representative or "best" grid point is selected by looking for the highest Spearman correlation between CAMS forecasts and measurements among all grid points that are within 0.6 degrees of latitude and 0.6 degrees of longitude (i.e., 1.5 times the grid resolution) from the station. This procedure is particularly useful for stations
located at the coast or in mountainous regions, i.e. where horizontal gradients of most variables are very large. For 3D variables, the best grid point is selected from the two closest pressure levels in a standard atmosphere. Moreover, 3D variables are linearly interpolated to hourly intervals from the original 3-hourly resolution.

Each ET model is trained online whenever it is needed, using all data that are not part of the test period required by the user. In the typical case, the user submits recent data and all the previous data are used for training. However,
the analysis of older data is also possible: in that case, the training period can include some data that were measured after the test period.

We performed an initial subjective feature selection of 20 variables from the CAMS forecasts, based on expert knowledge. Of these, only the 13 listed in Table 1 were kept after testing their impact on the model performance. These variables are complemented by timestamp, day of year and time of day, for a total of 16 model features.
However, we only use a subset (up to 12 features) for each gas species that we want to predict.

We evaluate the impact of each feature by calculating the permutation importance, which is defined as the decrease in model performance when a feature is randomly shuffled. The permutation approach has the advantage that it can make use of unseen data that were not used for training, hence avoiding the overestimation of the importance due to overfitting. On the other hand, the importance of a feature that covariates with another can be
underestimated.

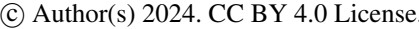

When choosing the features, we took into account cross correlations and avoided features that are strongly correlated with others. This led to the exclusion of pressure when modelling $CH_4$, because of its high correlation (r > 0.7 on average) with the total column quantity of $CH_4$. Although correlations were only analysed as network-wide averages, importances were analysed for each station that had at least 19 months of data since 2020 (of which at least 18 months were used for training). Therefore, features that have a low importance on average but a high importance for a particular station or gas species were still used.

The CAMS forecasts are downloaded after the end of each day from the Copernicus Atmosphere Data Store and immediately integrated into the GAW-QC database. Thus, CAMS+ is typically available with a one-day delay.

## 3.3 Monthly outliers with SARIMA

Monthly averages are less affected by noise and can be predicted with sufficient accuracy using classic time series analysis methods. Moreover, a few GAW stations only deliver monthly data, for which the two algorithms described so far cannot be used.

The SARIMA model is a popular method to extrapolate a data series that exhibits a seasonal cycle and a trend (e.g., Dabral and Murry, 2017; Cujia et al., 2019). A SARIMA model is defined by four components: an autoregression, meaning that the forecast is based on the previous observed values; a so-called "moving average", which is a regression on past prediction errors; a data integration, meaning that differences between time steps are considered rather than the absolute values, in order to make a time series more stationary (a requirement of SARIMA); and a seasonal component that consists of the previous three terms translated to the seasonal scale.

These components can be represented by 7 hyper-parameters, written as $(p, d, q)(P, D, Q)_m$, where $p$ is the autoregression order (i.e., the number of time lags), $d$ the degree of differencing (i.e., the number of times the data have had past values subtracted), and $q$ the moving average order. The capital letters indicate the same hyper-parameters for the seasonal component, with $m$ the period of a season (i.e., 12 months).

To find the best hyper-parameters, we first applied a pre-selection based on the physical and statistical properties of the data (Box-Jenkins approach). For example, $p$ was chosen equal to 1 to allow an influence from the previous month only, while $P$ was chosen equal to 0, implying that the system has no memory over a period of one year. The remaining hyper-parameters were selected through a grid search based on all available GAW series from 2015 onward, with at least 36 months of training data and 12 (consecutive) months of validation data. This led to the choice of a $(1,0,0)(0,1,1)_{12}$ model. Note that both the integration and the moving average components are only used



at the seasonal level. Even though this model is not the best one for each and every station and stationarity cannot

always be satisfied, we found that in a large majority of cases it delivers sufficiently accurate predictions for our

purposes.

We use the implementation of SARIMA available from the Python library statsmodels, which also provides the

possibility to add a deterministic trend function to the model. By default, we assume a linear trend (including for

the validation), but the user can change this setting.


## 3.4 Performance scores

### 3.4.1 Sub-LOF

The performance of the outlier detection is based on the ICOS flagged data (see Sect. 2). We use the Area Under

the Precision-Recall Curve (AUC-PR; Davis and Goadrich, 2006), a common metric that takes into account both the

ability of the algorithm to detect outliers and its tendency to produce false positives. These two properties are

represented by the Recall (or hit rate) and the Precision, respectively, which are defined as:

$$Recall = \frac{TP}{TP+FN} , \tag{1}$$

$$Precision = \frac{TP}{TP+FP} , \tag{2}$$

where TP are the true positives, FN the false negatives and FP the false positives. The Precision-Recall Curve is then

constructed by calculating Precision and Recall for a number of possible thresholds of the Sub-LOF score, above

which data points are considered outliers. Given that both Precision and Recall can have values between [0,1], the

AUC-PR score also covers the same range, with 1 representing the ideal scenario in which all outliers are detected

without false positives independently from the threshold. In real-world applications, any threshold entails a trade-

off between Recall and Precision, so that an AUC-PR score around 0.5 already represents an extremely good

performance (see, e.g., Schmidl et al., 2022). We use the algorithm implemented in the scikit-learn library, based on

the trapezoidal rule, to calculate the AUC. Figure 2 shows an example of a PR curve: in this case, to obtain 100%

Precision (i.e., no false positives) we would need to sacrifice over 80% of the maximum Recall.

The classic AUC-PR score is particularly suitable to evaluate the detection of individual, isolated outliers. However,

outliers in time series often occur as sequences. It is then usually not necessary to detect every single outlier, but it

is sufficient to detect at least one outlier per sequence. In order to tackle this issue, Tatbul et al. (2018) have





developed range-based Recall ($R_T$) and Precision ($P_T$) that are generally more suitable to validate outlier detection in time series.

$R_T$ and $P_T$ are defined by three parameters: $\alpha$, $\gamma$ and $\delta$. $\alpha$ can range between [0,1] and represents the relative importance of rewarding existence (relevant only for $R_T$). In our case $\alpha=1$, meaning that even if a sequence of anomalies is detected at just one data point, it will reward the maximum score. $\gamma$ represents cardinality and is a function of the number of flagged sequences that overlap with a detected sequence (or vice versa, depending on which score is being calculated). We use the recommended value $\gamma(x)=1/x$. $\delta$ is a function that defines which part of the sequence has more importance; in our case $\delta$ is a constant, meaning that we do not care about which part of the sequence is detected. We use the implementation of $R_T$ and $P_T$ provided by the Python library PRTS.

The resulting AUC-$P_TR_T$ score is more suitable for our benchmark data set, where sequences of flagged data tend to be artificially too long. In particular, $R_T$ should be more meaningful than the classic Recall. However, we show both classic and range-based scores, also to facilitate the comparison with other algorithms. The example of Fig. 2 depicts a typical situation in which the AUC-$P_TR_T$ score is slightly higher than the AUC-PR score.

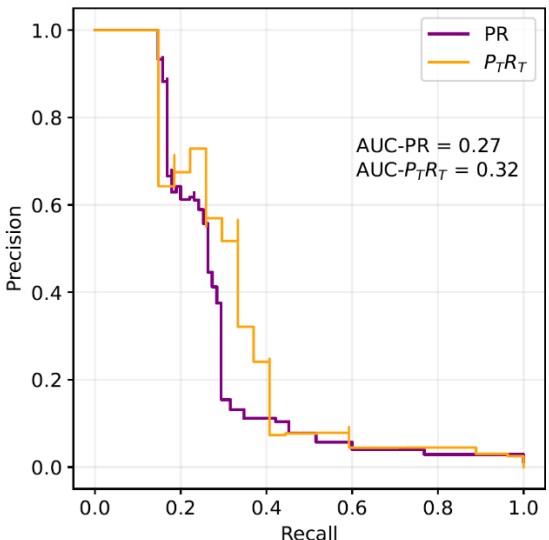

**Figure 2: Example of PR curve for a CO₂ series with 99 flags.**



### 3.4.2 CAMS+

The training of the ET model uses the Mean Squared Error (MSE) to measure the quality of a split. For grid search and validation we use the so-called Mean Squared Skill Score (MSSS), defined as:

$$MSSS = 1 - \frac{\text{MSE}_+}{\text{MSE}_c},$$ (3)

where $\text{MSE}_+$ and $\text{MSE}_c$ indicate the MSE of CAMS+ and of the original CAMS forecasts. The MSSS can assume values in the range $[-\infty, 1]$ and is larger than zero when CAMS+ has a lower (i.e., better) MSE than the CAMS forecasts. We also use the coefficient of determination ($R^2$), defined as:

$$R^2 = 1 - \frac{RSS}{TSS},$$ (4)

where RSS is the sum of squares of residuals and TSS is the total sum of squares. $R^2$ represents the ability of the model to perform better than a constant prediction with zero average bias, equivalent to $R^2=0$.

## 4 Validation

### 4.1 Sub-LOF

Figure 3 shows the distributions of the AUC-PR and AUC-$P_T R_T$ scores for different values of $n_s$ (with $k=100$) for $CH_4$,
CO and $CO_2$. The choice of the best value for $n_s$ depends on which score is considered: somewhat counterintuitively, the AUC-PR score favours a higher $n_s$ (i.e., longer sub-sequences) than the AUC-$P_T R_T$ score. On the other hand, differences between variables are rather small, although CO produces slightly lower scores than the other gases ($O_3$ could not be tested due to lack of suitable data). Our algorithm uses $n_s=3$ for all variables, a choice that maximizes the AUC-$P_T R_T$ without penalizing much the classic score. Through a similar exercise, we found a smaller influence
of $k$ on the scores (values between 30 and 300 were tested, corresponding to ca. 0.1% to 1% of the sample size; not shown).

A comparison with two alternative outlier detection algorithms (Sub-Isolation Forest, Liu et al., 2008; and PhaseSpace Support Vector Machines, Ma and Perkins, 2003) confirmed the good performance of Sub-LOF (Fig. S3).



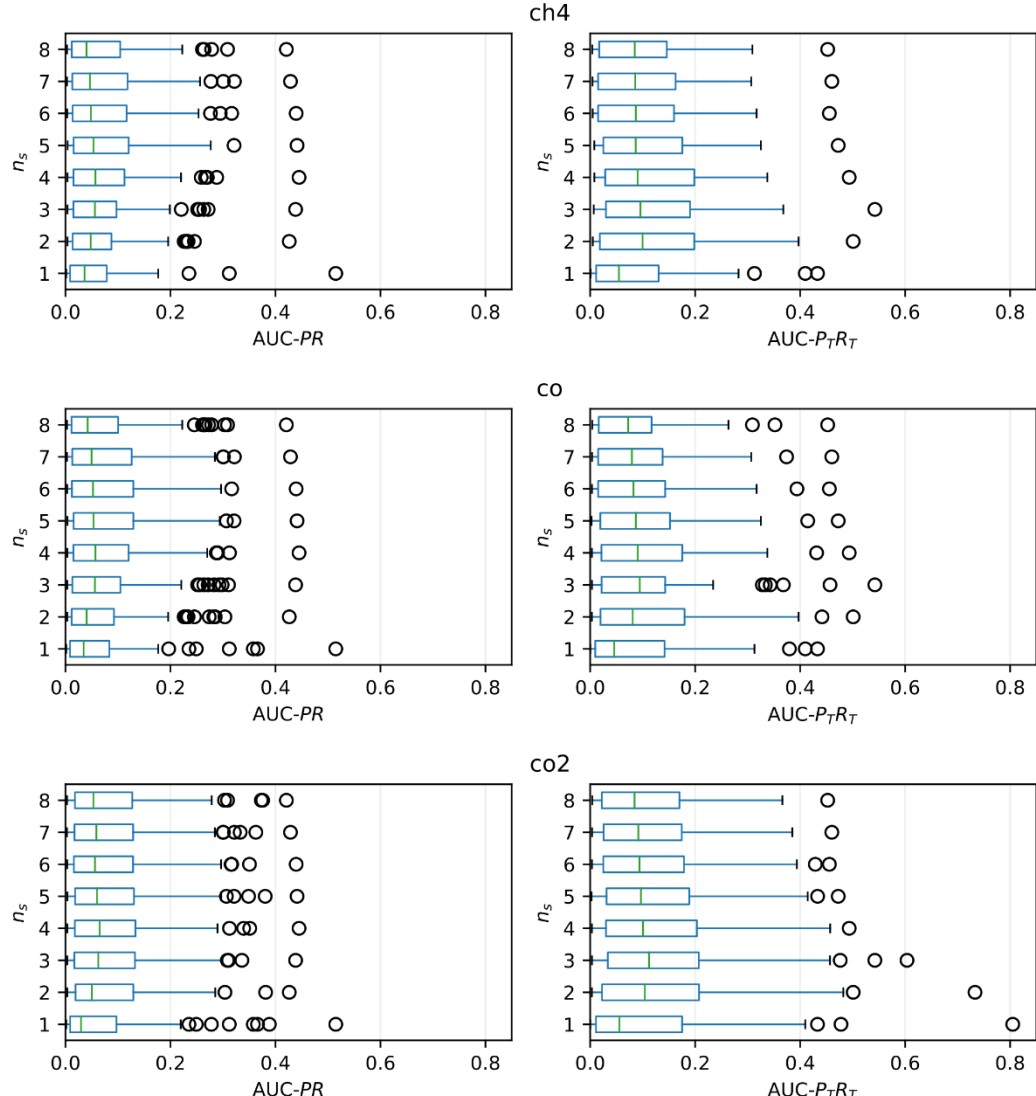


**Figure 3: Distribution of the AUC-PR (left) and ACU-$P_TR_T$ (right) scores of Sub-LOF for different values of the subsequence length ($n_s$; in hours) with $k$=100. Note that $n_s$=1 corresponds to a standard, univariate LOF algorithm. The vertical lines indicate the median of the tested stations, the boxes indicate the interquartile range (IQR), while the whiskers extend to up to 1.5 IQR from the edges of the boxes.**


## 4.2 CAMS+

We calculated feature importances for each year between 2020-2023 (using the remaining years for training) and then considered the maximum of the four resulting values (Fig. 4) to evaluate the usefulness of each feature. Where





the CAMS forecasts perform well, the respective gas mole fraction as simulated in the forecasts has by far the
highest importance. This is more often the case for CO and $O_3$, and less for $CH_4$. The reason is that no $CH_4$
measurements are assimilated in the CAMS forecasts.

The day of year (doy) is often the most important feature when the CAMS forecasts perform poorly, because the
seasonal cycle is the easiest source of variability to model. However, importances are very heterogeneous across
the stations. Some of the results also give interesting hints to the physical processes driving the concentration, in
particular the high importance of a certain wind direction for $CH_4$ at some stations.

The near absence of negative values (the lowest value is -0.0026) implies that there are no features that consistently
lead to overfitting at any station, although this can happen in individual years (see the minimum values in Fig. S4).
The trend feature (i.e., timestamp) has sometime a relatively large importance for $CH_4$: this may be related to a poor
representation of the trend in the CAMS data. In fact, this feature was added mainly to account for possible future
inhomogeneities in the CAMS forecasts.

The MSSS is positive for a large majority of tested stations in all years, that is, the ET model is usually able to improve
the CAMS forecasts (Fig. 5). Again, the variability across stations and gas species is large.

The CAMS forecasts generally underestimate $CH_4$, therefore simply correcting for a constant bias would in many
cases result in a MSSS close to 1. The CAMS forecasts are also generally poor at reproducing the variability of $CH_4$,
and this is often improved in CAMS+ thanks to the ancillary features that are fed to the ET model (see Fig. 4).

CO and $O_3$ are much better reproduced by the CAMS forecasts, making it more challenging for the ET model to
achieve positive MSSS. Nevertheless, negative MSSSs are very rare and usually affect elevated stations that are
predominantly exposed to the free troposphere, for which the original CAMS forecasts can already simulate
accurately the mole fraction of CO and $O_3$, leaving little room for improvement to the ET model. Even for $CH_4$ the
performance of the CAMS forecasts is notably better at these stations, as is evident from the importances in Fig. 4.

The year 2023 represents an interesting case for CO, with negative MSSS at seven stations out of 27. This was a
year characterised by an extremely active wildfire season in North America (Jones et al., 2024), which strongly
affected CO concentrations also in Europe (Byrne et al., 2024) and has no analogues in our relatively short training
period.

The $R^2$ score provides a better metric to compare the performance across variables, because it is not defined by the
performance of the CAMS forecasts. Nevertheless, the poor performance of the CAMS forecasts for $CH_4$ is the
ultimate reason for the low $R^2$ scores for that variable.




**Figure 4: Maximum feature importance for the three available gas species (CO, O₃, CH₄) at validation stations (identified by the GAW code and ordered by decreasing latitude), based on 2020-2023 data and the MSSS. The feature importance is calculated separately for each year and then the maximum of the yearly values is taken (at least 3 years required). Note that values can be larger than 1. For tower stations the highest sampling height is used. For station names and coordinates see Table S1.**





The variable that scores best is $O_3$, in part because it exhibits larger and more predictable diurnal and seasonal cycles than CO. Moreover, the large outliers that can affect CO are especially detrimental when evaluating scores that rely on the MSE, as the case of 2023 also shows.

It is important to point out that a negative MSSS or $R^2$ score does not necessarily imply a bad performance of CAMS+. Many of the outliers in Fig. 5 are actually related to problems in the measurements that were not flagged

by the station operators, confirming that there is a need for better QC in the GAW community.

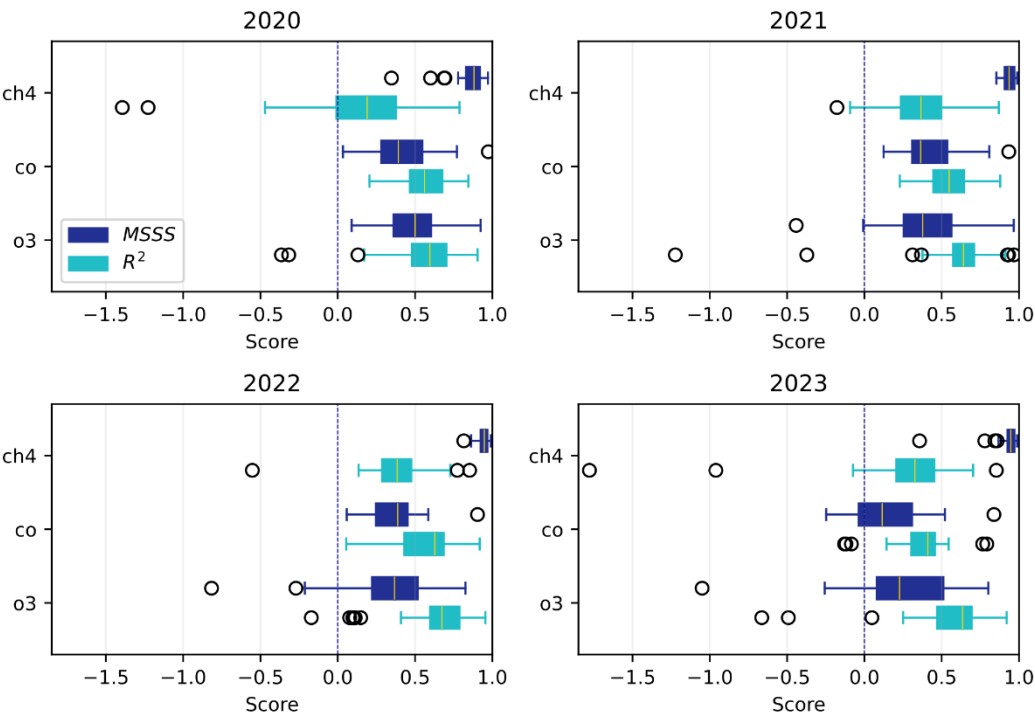

**Figure 5: Distribution of the MSSS and R² score by variable for each year between 2020-2023. The boxes are defined as in Fig. 3.**

**4.3 SARIMA**

To evaluate how accurately we can predict monthly means with SARIMA, we calculate absolute errors as a function of the prediction horizon, using 12 recent consecutive months for validation and the previous data (between 3 and 8 years) for training. The calculation is repeated for every possible 12-month period (January-December, February-




January, etc.), resulting in up to $12^2$ error values for each data series. This procedure allows us to minimise seasonal

influences from the results.

The resulting average errors (Fig. 6) are of ca. 10 ppb for $CH_4$ (i.e., ca. 0.5 of typical background values), 15 ppb for CO, 1.5 ppm for $CO_2$ (0.4%), and 3 ppb for $O_3$. However, the medians of the errors are lower.

As one would expect, the performance is best for a 1-month horizon and the absolute error increases with time. However, for CO and $CO_2$, the absolute error reaches a maximum after 6-8 months and then decreases again

slightly. The likely explanation for this behaviour is that the inter-annual variability of these gases has itself a large seasonal cycle, particularly at mid-latitudes (higher in winter for CO, in summer for $CO_2$). Therefore, a prediction starting from a season with low variability will tend to perform better on a 12-month than on a 6-month horizon, while a prediction starting from a season with high variability has high chances of performing poorly for all horizons.

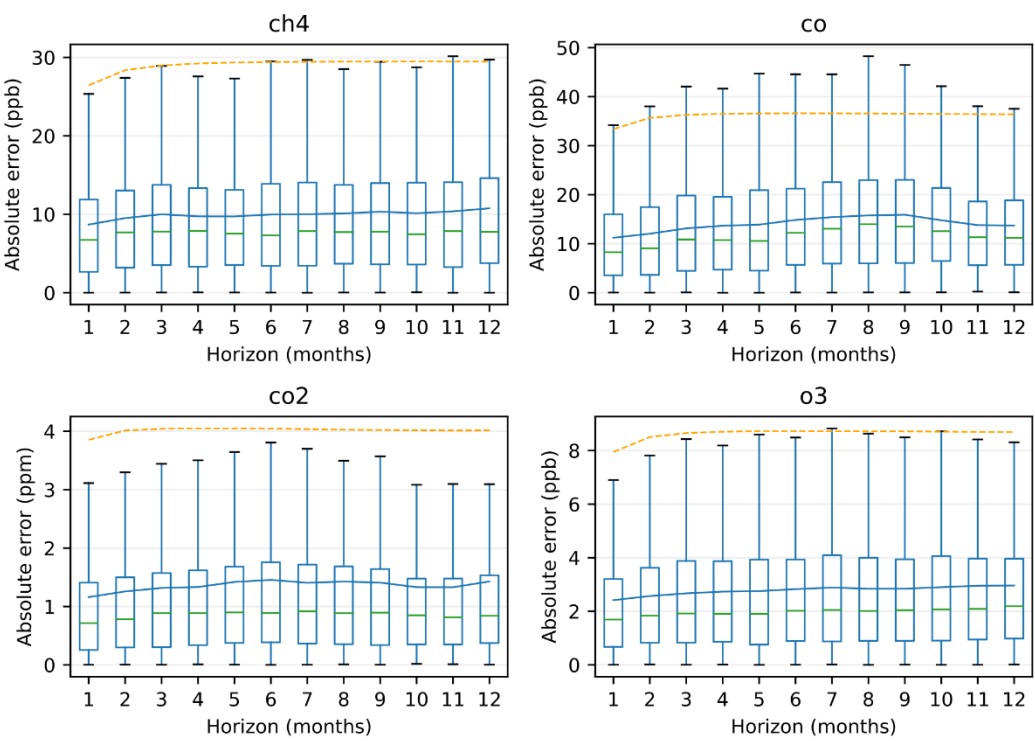


**Figure 6: Distribution of absolute prediction errors of the SARIMA model as a function of the prediction horizon. The blue line indicates the average of all tested stations. The orange dashed line shows half the average 99% confidence range produced by the SARIMA model. The boxes are defined as in Fig. 3, but outliers are not shown.**



**5 GAW-QC**

The algorithms described in the previous sections can be tested on new data from one of the supported GAW

stations through the web application GAW-QC (see www.empa.ch/gaw for the current link). Moreover, any period

of the historical data available in the GAW-QC database can be analysed.

Figure 7 gives a simple schematic of the data flow. GAW-QC is synchronized daily with the archive of the CAMS

forecasts, from which the relevant variables and grid points are extracted. Synchronization with the GAW world data

centres occurs at irregular intervals, although an extension to an automatic loading process is technically feasible

provided the data are available through appropriate APIs.

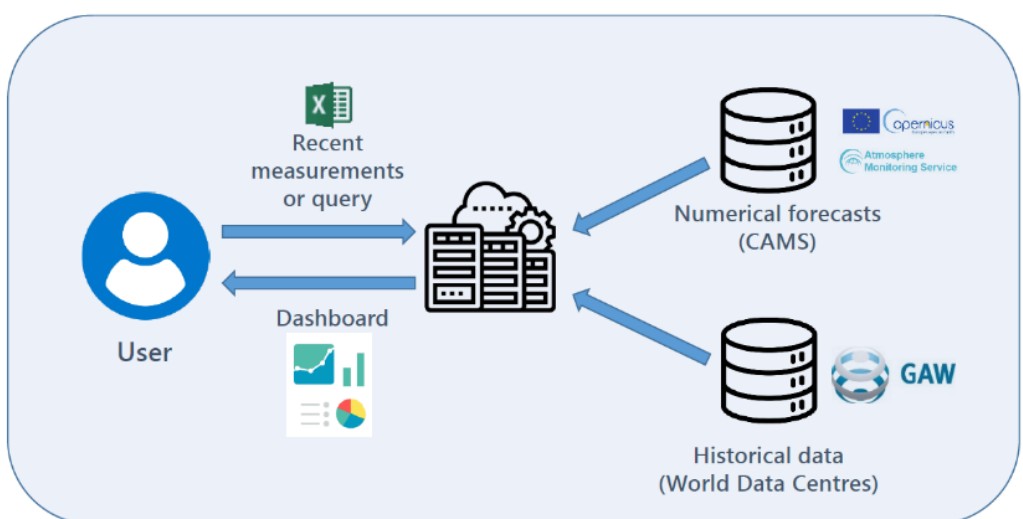

**Figure 7: Schematic diagram of the data flow in GAW-QC.**


To use GAW-QC, the user must select a station, a variable, and the sampling height. If new data are analysed, these

must be uploaded as a csv or xls/xlsx file. Otherwise, an available target period of up to one year must be selected

(data availability depends on the station and variable).

GAW-QC consists of an interactive dashboard with three panels: a panel for hourly data, a panel for monthly data,

and a panel for additional visual checks (Fig. 8). The data shown in each panel can be downloaded through an

export button, including the information on flags. A user guide is provided in the form of a wiki, where a tutorial





can also be downloaded in pdf format. Moreover, a short explanation of the dashboard functionalities is quickly accessible through help buttons on each panel.

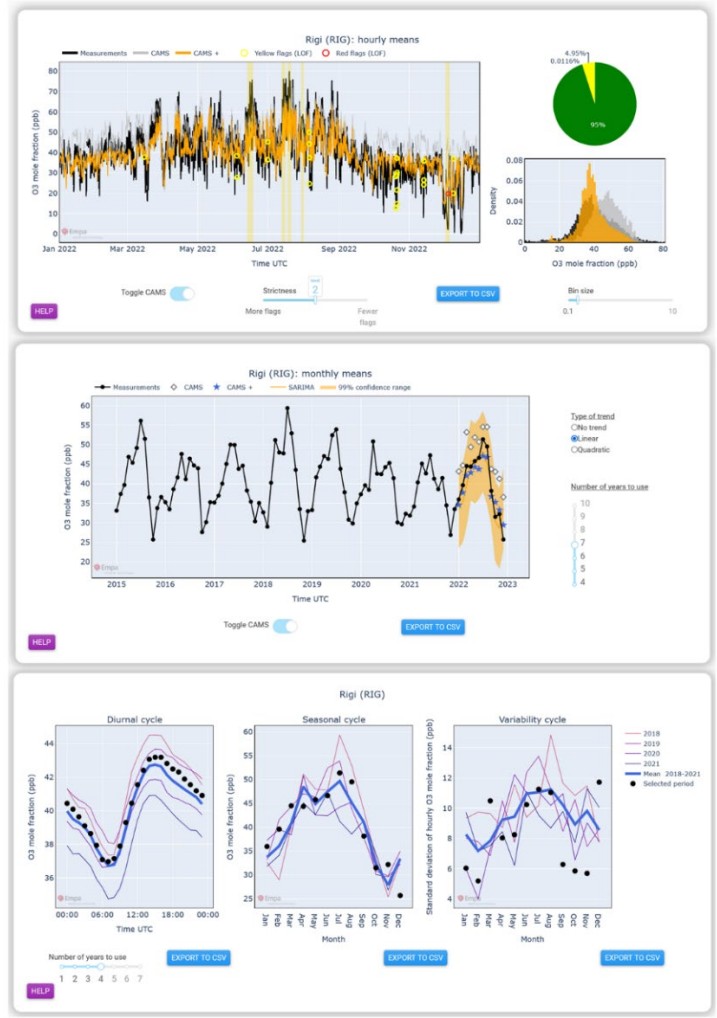

**Figure 8: Example of a GAW-QC dashboard.**

**5.1 Hourly data**

The hourly data panel (Fig.8 top) shows the measurements, the CAMS forecasts and CAMS+ for the selected station. Data points that are flagged by Sub-LOF are indicated by yellow and red circles, which represent the exceedance of two different threshold levels by the LOF anomaly score. By default, yellow flags are triggered when exceeding

the 99.6th percentile of the anomaly scores of the historical data (i.e., the data of the analysed station that are outside the target period). The threshold can be adjusted by the user to the 99.3th or 99.9th percentile through a



"strictness" slider located below the plot. Red flags are triggered when the anomaly score is double the threshold or higher. By hovering over a flag, the time and value of the affected measurement appears.

In addition, periods with significant differences between the measurements and CAMS+ are highlighted by a yellow shading. Again, the definition of "significant" depends on a threshold based on historical data that the user can adjust. For this we apply a 50-hour running median filter to the series of differences between measurements and CAMS+. The output of the filter is equivalent to an anomaly score. In case of missing values, at least 50% of valid data is required within the 50-hour window to calculate the score. Initially the threshold is set at twice the 99th percentile of the historical data and can be increased or decreased by one percentile through the strictness slider. If any data point exceeds the threshold, then the whole 50-hour window is highlighted, that is, all the points in that window receive a yellow flag. Yellow flags by Sub-LOF are upgraded to red flags if the affected data point is within a highlighted region.

On the right-hand side of the panel, two additional plots show the quantity of flagged data on a pie chart and the distribution of the data on a histogram. The latter is useful to spot systematic biases and to evaluate the ability of CAMS+ to reproduce extremes.

### 5.2 Monthly data

The second panel (Fig.8 middle) shows the evolution of the observed monthly means for up to 10 years before the target period, where the number of years can be adjusted by the user through a slider on the right-hand side of the panel. This number also determines which data are fed to the SARIMA model. Therefore, changing the number of years will change the SARIMA prediction accordingly.

In the target period three different predictions are shown: the CAMS forecasts, CAMS+ and SARIMA, the latter including its confidence range (orange shading). Monthly means of measurements that are outside the SARIMA confidence range (see also Fig. 6) are flagged, while no graphical highlight is used for differences with CAMS+.

The SARIMA model that we employ requires to input a deterministic long-term trend. By default, a linear trend is assumed, which is a good approximation in most cases for the maximum 10 years of consideration. The user can change this setting to "no trend", which might be a better choice for CO and $O_3$ at some stations, or to a quadratic trend, which might be suitable for $CH_4$ in some periods.



## 5.3 Visual QC

The last panel (Fig.8 bottom) provides three plots where different statistics calculated over the target period are
compared with other years and with their average. The statistics are the diurnal cycle, the seasonal cycle of the
monthly mean values and the seasonal cycle of the variability. The variability is defined as the standard deviation
of the hourly measurements over a month.

The user can choose the number of years to plot through the slider in the bottom-left corner. The target period is
represented by thick black dots. The interpretation of this panel is purely subjective, as neither predictions nor flags
are provided. Nevertheless, these visual comparisons can be useful to interpret the flags of the previous panels and
can occasionally facilitate the detection of some types of issues, for examples mistakes in the time zone conversion.

## 5.4 Case study

To showcase how GAW-QC deals with real world measurements, we show in Fig. 9 the dashboard for the raw CO
measurements made at the station of Jungfraujoch between April and June 2024. The time series is characterised
by two sudden drops in CO mole fraction: a first, short one in mid-April, and a second, longer and noisier one at
the end of May.

The first drop in CO, which lasted between 12 and 15 April 2024, represents a real atmospheric phenomenon,
possibly caused by stratospheric intrusion (Esler et al., 2001; Cui et al., 2009), and is relatively well reproduced by
CAMS+. Hence, this period is correctly not highlighted by GAW-QC as anomalous. The second drop, on the other
hand, causes a significant difference between measurements and CAMS+; moreover, several data points in that
period are flagged by Sub-LOF. An inspection at the station revealed that between 23 May and 5 June, the inlet
had been buried under an exceptionally deep snow layer, which caused an underestimation of CO. In this case, the
operator should flag all observations in the affected period as incorrect.

Note that, to improve readability, periods with significant differences between measurements and CAMS+ that are
less than 50 hours apart are merged together into a single shaded area on the plot (but not in the export file).
Because the difference was not significant for a brief period between 31 May and 2 June, some flags within the
shaded area are yellow and not red.

Given the relatively large inter-annual variability of CO and the short duration of the event, this issue is not clearly
visible on the other panels of the dashboard.






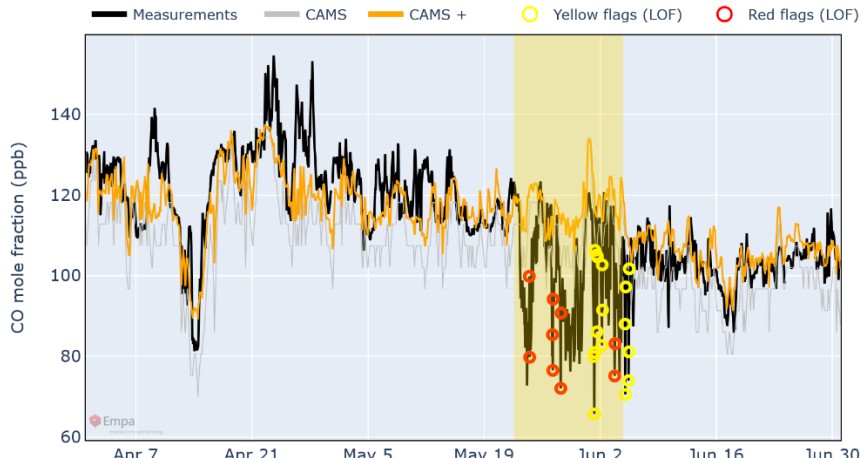

**Figure 9: Hourly plot produced by GAW-QC for raw CO data measured at the station of Jungfraujoch between April and June 2024.**

## 6 Concluding remarks

GAW-QC is a web application that helps GAW station operators and GAW data users to spot quality issues in atmospheric concentration data series through an interactive dashboard. The application brings together modern anomaly detection algorithms, numerical forecasts, and user expertise to achieve high quality data series of the most commonly measured trace gases. Depending on the success of the application and user feedback, additional gas species may be added in the future.

The flags provided by GAW-QC are intended as a guidance to expert decision and not as automated QC. In other words, it is still the station operator that must decide which measurements to flag as only the station operators have best local knowledge and access to all available information such as logbooks and maintenance reports. Indeed, it is important to make full use of existing metadata and local expertise that cannot be automatically incorporated into the GAW-QC algorithms. In fact, some of the automatically flagged values may be correct

measurements of rare phenomena that should be retained in the time series.

A remaining issue in the use of numerical forecasts is the potential vulnerability to future major upgrades of the CAMS forecasts of the machine learning model that we use for downscaling. For example, an increase in resolution of the CAMS forecasts would likely result in a poor performance of CAMS+, because it would have been trained on lower resolution data. However, this would not strongly impact the overall functionality of the tool. Note that

already in the current version we do not provide CAMS+ forecasts for $CO_2$ due to the lack of suitable CAMS

products, although the release of a new CAMS forecast product focusing on greenhouse gases in October 2024 will probably allow the addition of $CO_2$ forecasts in the near future.

The goals of GAW-QC are manifold: (i) to provide near-realtime QC, allowing station operators to timely detect and deal with instrumentation malfunctions or sources of contamination, thus avoiding the loss of months or years of
data, (ii) to promote a correct and consistent QC procedure worldwide, (iii) to help scientists to evaluate the quality of historical series, hence improving the quality of their research, (iv) to motivate the station operators to regularly submit their data to the world data centres; and, finally, (v) to support training activities within the framework of the GAW programme.

**Data availability.** GAW data for $CH_4$, CO and $CO_2$ are available at the World Data Centre for Greenhouse Gases (WDCGG; https://gaw.kishou.go.jp). GAW data for $O_3$ are available at the World Data Centre for Reactive Gases (WDCRG; https://ebas-data.nilu.no/). ICOS data are available on the ICOS Carbon Portal (https://data.icos-cp.eu). The CAMS forecasts are available on the Copernicus Atmosphere Data Store (https://ads.atmosphere.copernicus.eu). The relevant GAW and CAMS data can also be downloaded directly from
GAW-QC using the export function.

**Author contributions.** YB: Conceptualization, Methodology, Software, Visualization, Writing – original draft preparation, Writing – review & editing. MS: Conceptualization, Supervision, Writing – review & editing. SB: Software, Writing – review & editing. LE: Supervision, Writing – review & editing.

**Competing interests.** The authors declare that they have no conflict of interest.

**Acknowledgments.** This work was supported by the Federal Office for Meteorology and Climatology MeteoSwiss through engagement in the Global Atmosphere Watch programme. We thank Carl Remlinger and the Swiss Data
Science Center (SDSC) for many useful discussions and suggestions. The CAMS forecasts were provided by the Copernicus Atmosphere Monitoring Service (https://atmosphere.copernicus.eu).



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
