# Peer review of "Technical note: An interactive dashboard to facilitate quality control of in-situ atmospheric composition measurements"

_EGUsphere, 2024_

## Author Comment (AC1)

Materials Science and Technology

We are grateful to both referees for their constructive and encouraging feedback. Since the manuscript was submitted, the dashboard has undergone several updates that already incorporate some of their suggestions, including the addition of two new variables ($CO_2$ and $N_2O$). In the following, we answer the reviews point by point (answers in blue).

**Referee 1**

This manuscript introduced an interactive dashboard tool for detecting anomalous measurements which may help identifying measurement quality issue. This tool is based on three independent algorithms. The algorithms were discussed in the manuscript with application in actual data. The authors correctly pointed out that the anomalous points may or may not be related to measurement problems and the measurement problems should be determined by the instrument scientists or operators. This reviewer believes this tool may help improve measurement quality, but the manuscript did not adequately discuss the effectiveness of this tool. One obvious issue is that not all measurement quality problems are shown as anomalies. The authors should revise the manuscript after considering the comments below.

**Major Comments:**

1.  The manuscript primarily focused on discussions about ways to detect anomalous data points. As stated by the authors, the anomalies may not be related to the measurement quality issues. In fact, many unexpected changes in data can be related to swift meteorological condition changes and/or "accidental" emission sources. In general data should not be flagged unless potential specific instrument and/or sampling issues are identified. In this context, it would be helpful for the non-data science readers to see the effectiveness of this tool to identify the measurement problems. This reviewer would like to see the percentage of the identified anomalous data that can be linked to measurement problems and what kind of measurement problems were identified using this tool. This reviewer believes it is an important factor for readers to evaluate the usefulness of the tool if they understand the underlying processes associated with the detected anomalous data.

    The manuscript already contains a systematic validation of the ability of Sub-LOF to detect real measurement problems, based on ICOS data (Fig. 3). However, as mentioned in the text, most ICOS station operators are rather generous with quality flags and, therefore, there is a tendency to underestimate the hit rate of the algorithm. Moreover, we do not have information of why the data were flagged.

    The core challenge lies in the scarcity of datasets containing known quality issues, as such data are typically removed prior to publication. Therefore, we can only perform detailed validation on a very limited number of case studies (coming from stations that we operate or that we audited), such as the one shown in Fig. 9. To strengthen the evaluation, we have added two additional case studies,

although these are still far from giving a comprehensive validation of the tool's capabilities, which remains practically unfeasible.

2. It is not widely accepted that the model products can be used to judge measurement problems. This is especially problematic for comparison with ground site observations as the models do not capture local scale meteorology and small emission sources, while both can be reflected in the observations at a given time and location. This may be a more serious problem when dealing with more reactive species.

While the CAMS forecasts offer valuable insights, they do have limitations - even after applying machine learning corrections - and may not always be reliable, as implicitly illustrated in Fig. 5. It is ultimately up to the user to judge whether they can be trusted for the specific time series. In fact, there is a "toggle CAMS" switch in the app exactly for this reason, which allows one to hide/unhide the CAMS output. We emphasize this point more explicitly in the text. We agree with the reviewer that our approach is more challenging for reactive species with large spatial and temporal variability. This is one reason we focus on long-lived greenhouse gases and surface ozone, which typically exhibit relatively low spatial and temporal variability, especially at remote locations such as most GAW stations. Since the release of the discussion version, we added another long-lived greenhouse gas (nitrous oxide, N2O) to the GAW-QC portfolio.

3. The actual measurement problems are often discovered by examine the relationship between variables observed at the same location and time. For example, the stratospheric intrusion case presented in the manuscript should be characterized by higher ozone levels and lower water vapor, in addition to low CO levels. Looking an CO times alone with the model output cannot be considered as convincing... The model can often have a phase shift. It would be much more useful if the tool can take advantage of the simultaneous measurements at a given site.

We agree and have added the possibility to upload an additional variable (for example humidity or any other trace gas or aerosol parameter) that can be visually compared with the analyzed variable.

**Specific Comments:**

1. Table 1.: The authors should explain why and how total column CH4/CO/O3, black carbon, and water vapor would help explain surface observations. This reviewer believes that the readers deserve some physical explanations.

We agree and will add possible explanations.

2. Line 92: "exceeds 5%", how sensitive is this choice? Also, why only the "highest sampling height" is selected? Is there often a sharp gradient observed?

The choice of 5% is arbitrary and aims at reducing the problem of overflagging. We will add an assessment of the quantitative impact on the number of flags.

> We chose the highest inlet because it is preferred by many data users (in particular, to validate model results). Choosing another height would not have changed the results significantly; we will add illustrative results for the lowest heights in the Supplement.

3. Sub-LOF: the authors should provide more description on the general principle of this algorithm.

> Will do.

4. Section 3.2. The authors should discuss the performance of the CAMS model and the difference between the forecast mode and reanalysis mode.

> Will do.

5. Line 167: "based on expert knowledge" is not an acceptable justification. The choice of the variables should be explained and justified based on atmospheric sciences.

> The subjective pre-selection was necessary for efficiency reasons, but ultimately, we kept only those variables that improved the forecast of the target variable. The "expert knowledge" refers to expertise in atmospheric sciences; we will improve the wording.

6. Section 3.4.1.: The authors should clearly define true positive, false positive, and false negative. Some examples should be provided to explain these concepts.

> Will do.

7. Figure 3: what do the circles represent?

> They represent individual values outside the range of the whiskers. We will add this to the Figure's caption.

**Referee 2**

**General comment**

This technical note by Brugnara et al. illustrates a novel tool for facilitating the quality control (QC) of in-situ atmospheric composition measurements for several WMO/GAW sites. The tool, which is available through an interactive dashboard, is based on three different algorithms for the identification of anomalous measurements.

A general consideration is that the tool gives the possibility either: (i) to analyze already submitted data (e.g., hourly data available at the world data centers, that are already supposed to have been validated by the PIs), and to (ii) submit the user's own data for a quick check, e.g., before submitting data to the world data centers.

Regarding point (i), I would like the authors to clarify how they envision handling previously validated datasets that the GAW-QC tool subsequently flags as outliers. In such cases, what is the added value of GAW-QC? There is a risk of confusing users if data considered valid by the station PIs are flagged as outliers by GAW-QC.

[Figure]

We aim to make clear in the documentation (for example, on the landing page in the wiki) that outliers are not necessarily indicative of erroneous data (actually, in most cases they are not). One important added value of GAW-QC is to highlight anomalous events – regardless of the cause - which can be interesting for a wide range of users. Moreover, comparing these anomalies with CAMS products can be of value to understand the limitations in those forecasts. We have added one entry to our FAQ page that addresses specifically this issue.

For point (ii), have the authors considered making the tool integrable into possible existing automatic QC pipelines at WMO/GAW sites under examination? This could allow for seamless, automated usage (e.g., eliminating the need for PIs to manually upload datasets every few months). Such integration could significantly broaden adoption.

This is a development that we are highly interested in pursuing, but requires additional actors and resources. As with many IT projects, we feel that it's best to first ensure a resilient and widely accepted code, before its wide, operational deployment.

The tool is certainly promising and can be widely used within PIs and operators at WMO/GAW stations. The manuscript is overall well written, and I believe it can be published only after the authors better clarify the general use of the tool concerning already validated data, and after addressing the specific comments below.

**Specific comments**

1.  Line 25: "critical" can be removed.

    Will do

2.  Lines 38–40: this statement may be too general. Near-real-time QC often depends on the capabilities of the station PI, whose responsibility is to check and submit data, typically on an annual basis. Please rephrase to avoid overgeneralization..

    Will do

3.  Line 61: "October 24". This information, as well as that in Fig. 1, can be revised and upgraded in the final version of the manuscript, as many months have passed since October 2024.

    Yes, absolutely, there have been many updates since.

4.  Table 1: does the second column indicate the variable name used in CAMS? If so, please change the column name.

    Yes, will do.

5.  Line 80: remove "on".

    Here, "on over" should become "on more than".

6.  Line 85: it is not totally clear to me how the ICOS data are used for validating CAMS data. If the CAMS data are then used to be compared with the observations, how would you treat the GAW stations that are also ICOS sites?

    ICOS data are not used to validate CAMS, they are used to validate the outlier detection algorithms. There was a typo in this paragraph ("CAMS operators" instead of "ICOS operators") that probably caused the confusion.

7.  Line 102: consider presenting the three methods as a numbered list for readability.

    Will do.

    Line 110: very often, to validate or flag some measurements at a station, it is necessary to analyze the behavior of several other compounds (to indicate, e.g., problems in a common sampling head, influence of local pollution, stratospheric intrusion events, …). As you stated here, the GAW-QC tool does not give the user the possibility of such analysis (which then relies on the PIs' expertise after a "preliminary" screening with GAW-QC); have the authors thought of providing to the user the possibility of analyzing different variables (already included in GAW-QC) in parallel?

    We have added the possibility to upload an additional variable (any variable) that can be visually compared with the analyzed variable.

8.  Line 132: how to deal with missing data records (differently than a timestamp with a missing value, I mean that the data record is totally missing)? Does the input dataset need to be padded, so that all records are present?

    No, the software takes care of that.

9.  Line 167: "based on expert knowledge", can you provide more details on this selection?

    Will do (see also answer to Referee 1).

10. Line 219: can you provide the definitions (or examples) of TP, FN, and FP?

    Will do.

11. Line 230: by detecting only one outlier per sequence, would then be the responsibility of each PI to accept/select the entire sequence as outliers?

    This line refers to the definition of the validation score, it does not mean that we necessarily detect one outlier per sequence. What is likely to happen is that not all outliers in a sequence will be flagged, but it is then straightforward to "fill the gaps" for the user. For example, if in a sequence of 100 measurements 60 are flagged as outliers, the more logical choice is probably to consider all 100 measurements as outliers.

12. Line 237: recommended by who? Please specify the source.

    Will do.

13. 3 and Fig. 5: do the circles represent the outliers of the distribution? If so, please specify it in the captions.

    Yes, will do.

14. 4.2: this section is not totally clear to me: what do you mean by "feature importance"? How is it calculated, and what important information can be retrieved from Fig. 4?

    The feature importance is a common metric for machine learning models. We will add more information.

15. Lines 319–320: "problems in the measurements": how can you define/hypothesize that there were problems, given that we must rely on the evaluation of the PIs/station operators for already submitted data?

These are cases where there can be no doubts that the problems are related to the measurements. We will mention one or two examples of the type of problem. Even the best station operator can sometimes forget a flag.

16. Line 367: the format for the user's input data is specified in the GAW-QC website, but I would add a sentence here on the format to be used.
Will do.

17. Figure 8: colors differ from the web version of GAW-QC; please update the figure and text references accordingly (e.g., "orange shading" at Line 392).
Will do.

18. Line 372: change to "two times or higher than the threshold".
Will do.

19. Line 383: "flagged data", do you mean by considering any type of flag?
Yes, we will clarify that.

20. Line 385: in the example provided, the histograms suggest CAMS and CAMS+ overestimate measurements. Do the authors have an explanation? I have observed this behavior at multiple stations (marine, high-elevation, etc.) while testing GAW-QC.
This depends on the variable, for example for methane there is usually an underestimation. It is hard to attempt an explanation without a deep understanding of the CAMS products. However, CAMS+ should not in general have systematic biases (there is no clear indication of that in the example).

21. Line 406: what do you mean by this? Aren't displayed and used data all in UTC by default?
When uploading data, the user must indicate the timezone. The conversion to UTC depends on the correctness of that input.

22. Line 413: have you confirmed this event by the use of other co-located measurements (e.g., an increase in $O_3$, a decrease in RH), or ancillary variables (e.g., PV)?
Yes, we used co-located as well as nearby measurements.

23. Line 447: this sentence should be rephrased in the final version, as the products may have been upgraded since October 2024.
Of course, we have indeed added $CO_2$ as well as $N_2O$.

---

## Author Response (AR1)

We are grateful to both referees for their constructive and encouraging feedback. Since the manuscript was submitted, the dashboard has undergone several updates that already incorporate some of their suggestions, including the addition of two new variables ( $CO_2$  and  $N_2O$ ). In the following, we answer the reviews point by point (answers in blue).

**Referee 1**

This manuscript introduced an interactive dashboard tool for detecting anomalous measurements which may help identifying measurement quality issue. This tool is based on three independent algorithms. The algorithms were discussed in the manuscript with application in actual data. The authors correctly pointed out that the anomalous points may or may not be related to measurement problems and the measurement problems should be determined by the instrument scientists or operators. This reviewer believes this tool may help improve measurement quality, but the manuscript did not adequately discuss the effectiveness of this tool. One obvious issue is that not all measurement quality problems are shown as anomalies. The authors should revise the manuscript after considering the comments below.

**Major Comments:**

1. The manuscript primarily focused on discussions about ways to detect anomalous data points. As stated by the authors, the anomalies may not be related to the measurement quality issues. In fact, many unexpected changes in data can be related to swift meteorological condition changes and/or "accidental" emission sources. In general data should not be flagged unless potential specific instrument and/or sampling issues are identified. In this context, it would be helpful for the non-data science readers to see the effectiveness of this tool to identify the measurement problems. This reviewer would like to see the percentage of the identified anomalous data that can be linked to measurement problems and what kind of measurement problems were identified using this tool. This reviewer believes it is an important factor for readers to evaluate the usefulness of the tool if they understand the underlying processes associated with the detected anomalous data.

The manuscript already contains a systematic validation of the ability of Sub-LOF to detect real measurement problems, based on ICOS data (Fig. 3). However, as mentioned in the text, most ICOS station operators are rather generous with quality flags and, therefore, there is a tendency to underestimate the hit rate of the algorithm. Moreover, we do not have information of why the data were flagged.

The core challenge lies in the scarcity of datasets containing known quality issues, as such data are typically removed prior to publication. Therefore, we can only perform detailed validation on a very limited number of case studies (coming from stations that we operate or that we audited), such as the one shown in Fig. 9. To strengthen the evaluation, we have added two additional case studies

- (Fig. 10 and 11), although these are still far from giving a comprehensive validation of the tool's capabilities, which remains practically unfeasible.
- 2. It is not widely accepted that the model products can be used to judge measurement problems. This is especially problematic for comparison with ground site observations as the models do not capture local scale meteorology and small emission sources, while both can be reflected in the observations at a given time and location. This may be a more serious problem when dealing with more reactive species.

While the CAMS forecasts offer valuable insights, they do have limitations - even after applying machine learning corrections - and may not always be reliable, as implicitly illustrated in Fig. 5. It is ultimately up to the user to judge whether they can be trusted for the specific time series. In fact, there is a "toggle CAMS" switch in the app exactly for this reason, which allows one to hide/unhide the CAMS output. We emphasize this point more explicitly in the text (end of Sect. 4.2) and we show one example (Fig. 11) where CAMS is not reliable.

We agree with the reviewer that our approach is more challenging for reactive species with large spatial and temporal variability. This is one reason we focus on long-lived greenhouse gases and surface ozone, which typically exhibit relatively low spatial and temporal variability, especially at remote locations such as most GAW stations. Since the release of the discussion version, we added another long-lived greenhouse gas (nitrous oxide, N2O) to the GAW-QC portfolio.

3. The actual measurement problems are often discovered by examine the relationship between variables observed at the same location and time. For example, the stratospheric intrusion case presented in the manuscript should be characterized by higher ozone levels and lower water vapor, in addition to low CO levels. Looking an CO times alone with the model output cannot be considered as convincing... The model can often have a phase shift. It would be much more useful if the tool can take advantage of the simultaneous measurements at a given site.

We agree and have added the possibility to upload an additional variable (for example humidity or any other trace gas or aerosol parameter) that can be visually compared with the analyzed variable. This is now mentioned in Sect. 5.1.

**Specific Comments:**

- 1. Table 1.: The authors should explain why and how total column CH4/CO/O3, black carbon, and water vapor would help explain surface observations. This reviewer believes that the readers deserve some physical explanations.
  - We agree and added possible explanations in Sect. 4.2.
- 2. Line 92: "exceeds 5%", how sensitive is this choice? Also, why only the "highest sampling height" is selected? Is there often a sharp gradient observed?
  - The choice of 5% is arbitrary and aims at reducing the problem of overflagging. We added an assessment of the quantitative impact on the number of flags in Sect. 2.

We chose the highest inlet because it is preferred by many data users (in particular, to validate model results). Choosing another height would not have changed the results significantly; we added illustrative results for the lowest heights in the Supplement (Fig. S4).

- 3. Sub-LOF: the authors should provide more description on the general principle of this algorithm.

  Done.
- 4. Section 3.2. The authors should discuss the performance of the CAMS model and the difference between the forecast mode and reanalysis mode.

Done.

- 5. Line 167: "based on expert knowledge" is not an acceptable justification. The choice of the variables should be explained and justified based on atmospheric sciences.
  - The subjective pre-selection was necessary for efficiency reasons, but ultimately, we kept only those variables that improved the forecast of the target variable. The "expert knowledge" refers to expertise in atmospheric sciences; we added the following sentence: "These are variables that we expected to have some correlation with the target variable due to known physical and chemical processes".
- 6. Section 3.4.1.: The authors should clearly define true positive, false positive, and false negative. Some examples should be provided to explain these concepts.

Done.

7. Figure 3: what do the circles represent?

They represent individual values outside the range of the whiskers. We added this to the Figure's caption.

**Referee 2**

**General comment**

This technical note by Brugnara et al. illustrates a novel tool for facilitating the quality control (QC) of in-situ atmospheric composition measurements for several WMO/GAW sites. The tool, which is available through an interactive dashboard, is based on three different algorithms for the identification of anomalous measurements.

A general consideration is that the tool gives the possibility either: (i) to analyze already submitted data (e.g., hourly data available at the world data centers, that are already supposed to have been validated by the Pls), and to (ii) submit the user's own data for a quick check, e.g., before submitting data to the world data centers.

Regarding point (i), I would like the authors to clarify how they envision handling previously validated datasets that the GAW-QC tool subsequently flags as outliers. In such cases, what is the added value of GAW-

QC? There is a risk of confusing users if data considered valid by the station PIs are flagged as outliers by GAW-QC.

We aim to make clear in the documentation (for example, on the landing page in the wiki) that outliers are not necessarily indicative of erroneous data (actually, in most cases they are not). One important added value of GAW-QC is to highlight anomalous events – regardless of the cause - which can be interesting for a wide range of users. Moreover, comparing these anomalies with CAMS products can be of value to understand the limitations in those forecasts. We have added one entry to our FAQ page (https://github.com/ybrugnara/gaw-qc/wiki/FAQ) that addresses specifically this issue.

For point (ii), have the authors considered making the tool integrable into possible existing automatic QC pipelines at WMO/GAW sites under examination? This could allow for seamless, automated usage (e.g., eliminating the need for PIs to manually upload datasets every few months). Such integration could significantly broaden adoption.

This is a development that we are highly interested in pursuing, but requires additional actors and resources. As with many IT projects, we feel that it's best to first ensure a resilient and widely accepted code, before its wide, operational deployment.

The tool is certainly promising and can be widely used within PIs and operators at WMO/GAW stations. The manuscript is overall well written, and I believe it can be published only after the authors better clarify the general use of the tool concerning already validated data, and after addressing the specific comments below.

**Specific comments**

- 1. Line 25: "critical" can be removed.
- 2. Lines 38–40: this statement may be too general. Near-real-time QC often depends on the capabilities of the station PI, whose responsibility is to check and submit data, typically on an annual basis. Please rephrase to avoid overgeneralization..

Done.

- 3. Line 61: "October 24". This information, as well as that in Fig. 1, can be revised and upgraded in the final version of the manuscript, as many months have passed since October 2024.

  Yes, absolutely, there have been many updates since. The information (including Fig. 1 and Table S1) is now updated to August 2025.
- 4. Table 1: does the second column indicate the variable name used in CAMS? If so, please change the column name.

Yes, done.

- Line 80: remove "on".We changed "on over" with "for more than".
- 6. Line 85: it is not totally clear to me how the ICOS data are used for validating CAMS data. If the CAMS data are then used to be compared with the observations, how would you treat the GAW stations that

are also ICOS sites?

ICOS data are not used to validate CAMS, they are used to validate the outlier detection algorithms. There was a typo in this paragraph ("CAMS operators" instead of "ICOS operators") that probably caused the confusion. This has been corrected.

- 7. Line 102: consider presenting the three methods as a numbered list for readability.

  Done.
- 8. Line 110: very often, to validate or flag some measurements at a station, it is necessary to analyze the behavior of several other compounds (to indicate, e.g., problems in a common sampling head, influence of local pollution, stratospheric intrusion events, ...). As you stated here, the GAW-QC tool does not give the user the possibility of such analysis (which then relies on the PIs' expertise after a "preliminary" screening with GAW-QC); have the authors thought of providing to the user the possibility of analyzing different variables (already included in GAW-QC) in parallel?
  - We have added the possibility to upload an additional variable (any variable) that can be visually compared with the analyzed variable. This is now mentioned in Sect. 5.1.
- 9. Line 132: how to deal with missing data records (differently than a timestamp with a missing value, I mean that the data record is totally missing)? Does the input dataset need to be padded, so that all records are present?
  - No, the software takes care of that on the assumption that the data have hourly resolution.
- 10. Line 167: "based on expert knowledge", can you provide more details on this selection?

  Done (see also answer to Referee 1).
- 11. Line 219: can you provide the definitions (or examples) of TP, FN, and FP?

  Done.
- 12. Line 230: by detecting only one outlier per sequence, would then be the responsibility of each PI to accept/select the entire sequence as outliers?
  - This line refers to the definition of the validation score, it does not mean that we necessarily detect one outlier per sequence. What is likely to happen is that not all outliers in a sequence will be flagged, but it is then straightforward to "fill the gaps" for the user, especially with the aid of metadata. For example, if in a sequence of 100 measurements 60 are flagged as outliers, the more logical choice is probably to consider all 100 measurements as outliers.
- 13. Line 237: recommended by who? Please specify the source.

  Done.
- 14. 3 and Fig. 5: do the circles represent the outliers of the distribution? If so, please specify it in the captions.

Yes, done.

15. 4.2: this section is not totally clear to me: what do you mean by "feature importance"? How is it calculated, and what important information can be retrieved from Fig. 4?

The feature importance is a common metric for machine learning models and it is defined in Sect. 3.2. We added a reminder in Sect. 4.2.

- 16. Lines 319–320: "problems in the measurements": how can you define/hypothesize that there were problems, given that we must rely on the evaluation of the Pls/station operators for already submitted data? These are cases where there can be no doubts that the problems are related to the measurements. -We mentioned typical examples, but we did not want to point fingers at specific stations.
- 17. Line 367: the format for the user's input data is specified in the GAW-QC website, but I would add a sentence here on the format to be used.

Done.

- 18. Figure 8: colors differ from the web version of GAW-QC; please update the figure and text references accordingly (e.g., "orange shading" at Line 392).

  Done.
- 19. Line 372: change to "two times or higher than the threshold".
- 20. Line 383: "flagged data", do you mean by considering any type of flag? Yes, we clarified that.
- 21. Line 385: in the example provided, the histograms suggest CAMS and CAMS+ overestimate measurements. Do the authors have an explanation? I have observed this behavior at multiple stations (marine, high-elevation, etc.) while testing GAW-QC.
  - This depends on the variable, for example for methane there is usually an underestimation (see Fig. 10). It is hard to attempt an explanation without a deep understanding of the CAMS products. Biases can vary strongly among different variables and regions (see latest quality assessment report by CAMS team: https://atmosphere.copernicus.eu/sites/default/files/publications/41\_CAMS2\_82\_2023SC2\_D82.1.1.14-DJF2025.pdf). However, CAMS+ should not in general have systematic biases (there is no clear indication of that in the example, nor did we find systematic biases in the validation data set).
- 22. Line 406: what do you mean by this? Aren't displayed and used data all in UTC by default?

  When uploading data, the user must indicate the timezone. The conversion to UTC depends on the correctness of that input.
- 23. Line 413: have you confirmed this event by the use of other co-located measurements (e.g., an increase in O3, a decrease in RH), or ancillary variables (e.g., PV)?

  Yes, we used co-located as well as nearby measurements.
- 24. Line 447: this sentence should be rephrased in the final version, as the products may have been upgraded since October 2024.
  - Of course, we have indeed added CO2 as well as N2O.